# MetaGAN: An Adversarial Approach to Few-Shot Learning

**Ruixiang Zhang**[*][†]
MILA, Université de Montréal
sodabeta7@gmail.com

**Tong Che**[*]
MILA, Université de Montréal
tongcheprivate@gmail.com

**Zoubin Ghahramani**
University of Cambridge
zoubin@cam.ac.uk

**Yoshua Bengio**
MILA, Université de Montréal, CIFAR Senior Fellow
yoshua.bengio@mila.quebec

**Yangqiu Song**
HKUST
yqsong@cse.ust.hk

## Abstract

In this paper, we propose a conceptually simple and general framework called MetaGAN for few-shot learning problems. Most state-of-the-art few-shot classification models can be integrated with MetaGAN in a principled and straightforward way. By introducing an adversarial generator conditioned on tasks, we augment vanilla few-shot classification models with the ability to discriminate between real and fake data. We argue that this GAN-based approach can help few-shot classifiers to learn sharper decision boundary, which could generalize better. We show that with our MetaGAN framework, we can extend supervised few-shot learning models to naturally cope with unlabeled data. Different from previous work in semi-supervised few-shot learning, our algorithms can deal with semi-supervision at both sample-level and task-level. We give theoretical justifications of the strength of MetaGAN, and validate the effectiveness of MetaGAN on challenging few-shot image classification benchmarks.

## 1  INTRODUCTION

Deep neural networks have achieved great success in many artificial intelligence tasks. However, they tend to struggle when data is scarce or when they need to adapt to new tasks within a few numbers of steps. On the other hand, humans are able to learn new concepts quickly, given just a few examples. The reason for this performance gap between human and artificial learners is usually explained as that humans can effectively utilize prior experiences and knowledge when learning a new task, while artificial learners usually seriously overfit without the necessary prior knowledge.

Meta-learning [Thrun, 1998, Hochreiter et al., 2001] addresses this problem by training a particular adaptation strategy to a distribution of similar tasks, trying to extract transferable patterns useful for many tasks. Recently, many different meta-learning or few-shot learning algorithms have been proposed. These algorithms may take the forms of learning a shared metric [Sung et al., 2018, Snell et al., 2017], a shared initialization of network parameters [Finn et al., 2017], shared optimization algorithms [Ravi and Larochelle, 2017, Munkhdalai et al., 2017, Munkhdalai and Yu, 2017], or generic inference networks [Santoro et al., 2016, Mishra et al., 2018] . In the context of few-shot classification, these algorithms try to learn a good strategy to form a correct decision boundary between different classes from only a few samples of data in each class.

---

[*]Equal contribution.
[†]Work done at HKUST

In this work we present MetaGAN as a general and flexible framework for few-shot learning. Most state-of-the-art few-shot learning models can be integrated into MetaGAN seamlessly. While most few-shot learning models consider how to effectively utilize few labeled data in a supervised learning way, semi-supervised few-shot learning which is studied recently in [Ren et al., 2018] is proposed when unlabeled data are available. In this paper, we show that both supervised few-shot learning and semi-supervised few-shot learning can be unified naturally with our prpoposed MetaGAN framework. We can further extend the sample-level semi-supervised learning proposed in [Ren et al., 2018] to the task level. For sample-level semi-supervised few-shot learning, we allow some training samples to be unlabeled within a task. These training samples can either come from the same classes as the labeled samples, or come from different "distractor" classes. For task-level semi-supervised few-shot learning, we also allow purely unsupervised tasks, in which both support and query samples are all unlabeled. Task-level semi-supervised few-shot learning can be very natural in practice. For example, we can have robots with cameras collecting data in different places. It is safe to assume that the data collected by one robot in a short time range come from a specific distribution, so classifying these images can be viewed as one task. But these tasks are completely unlabeled, both in the support and in the query sets. The MetaGAN algorithm is able to learn to infer the shape and boundaries of data manifolds of the task-specific data distribution from both labeled and unlabeled examples.

We provide both intuitive and formal theoretical justifications on the key idea behind MetaGAN. The main difficulty in few-shot learning is how to form generalizable decision boundaries from a small number of training samples. We argue that adversarial training can help few-shot learning models by making it easier to learn better decision boundaries between different classes. Although training data is usually very limited for each task, we show that how fake data generated by a non-perfect generator in MetaGAN can help the classifier identify much tighter decision boundaries (real-fake decision boundaries) and thus can help boost the performance of few-shot learning.

We demonstrate the effectiveness of MetaGAN on popular few-shot image classification benchmarks in both supervised and semi-supervised settings. We choose two representative few-shot learning models, MAML[Finn et al., 2017] representing models that learn to adapt using gradients, and Relation Network[Sung et al., 2018] representing models that learn distance metrics, and combine them with MetaGAN. [3] We show that MetaGAN can consistently improve the performance of popular few-shot classifiers in all of these scenarios.

## 2  BACKGROUND

### 2.1  FEW-SHOT LEARNING

We formally define few-shot learning problems as following: Given a distribution of tasks $P(\mathcal{T})$, a sample task $\mathcal{T}$ from $P(\mathcal{T})$ is given by a joint distribution $P_{X \times Y}^{\mathcal{T}}(\mathbf{x}, y)$, where the task is to predict $y$ given $\mathbf{x}$. We have a set of training sample tasks $\{\mathcal{T}_i\}_{i=1}^{N}$. Each training sample task $\mathcal{T}$ is a tuple $\mathcal{T} = (S_{\mathcal{T}}, Q_{\mathcal{T}})$, where the support set is denoted as $S_{\mathcal{T}} = S_{\mathcal{T}}^{s} \cup S_{\mathcal{T}}^{u}$, and the query set is denoted as $Q_{\mathcal{T}} = Q_{\mathcal{T}}^{s} \cup Q_{\mathcal{T}}^{u}$. The supervised support set $S_{\mathcal{T}}^{s} = \{(\mathbf{x}_1, y_1), (\mathbf{x}_2, y_2), \cdots (\mathbf{x}_{N \times K}, y_{N \times K})\}$ contains $K$ labeled samples from each of the $N$ classes (this is usually known as $K$-shot $N$-way classification). The optional unlabeled support set $S_{\mathcal{T}}^{u} = \{\mathbf{x}_1, \mathbf{x}_2, \cdots \mathbf{x}_M\}$ contains unlabeled samples from the same set of $N$ classes, which can also be empty in purely supervised cases. $Q_{\mathcal{T}}^{s} = \{(\mathbf{x}_1, y_1), (\mathbf{x}_2, y_2), \cdots (\mathbf{x}_T, y_T)\}$ is the supervised query dataset. $Q_{\mathcal{T}}^{u} = \{\mathbf{x}_1, \mathbf{x}_2, \cdots \mathbf{x}_P\}$ is the optional unlabeled query dataset. The objective of the model is to minimize the loss of its predictions on a query set, given the support set as input.

### 2.2  ADVERSARIAL TRAINING

The generative adversarial networks [Goodfellow et al., 2014] framework is one of the most popular approaches to generative modeling. It tries to adversarially train two neural networks, a generator and a discriminator. Adversarial training has seen a vast range of applications in recent years, such as semi-supervised learning [Dai et al., 2017, Salimans et al., 2016], unsupervised representation learning [Chen et al., 2016], imitation learning [Ho and Ermon, 2016] etc. However, few works have successfully combined adversarial training with few-shot learning. [Antoniou et al., 2018] proposed

to train a class conditioned GAN (DAGAN) to perform data augmentation. This is related to our proposal but is different in two aspects. 1) Their GAN model is trained separately from the classifier, only to provide additional data. 2) They treat generated data as real training data of the conditioned class. There are two drawbacks of this approach. First, GANs still have trouble in generating realistic samples in complex datasets such as ImageNet, so treating the generated images as real data in these datasets is questionable. Second, DAGAN can very easily run into mode collapsing. In many cases it is easy to collapse to an identity function — it just reconstruct the input image. Our approach does not require the generator to be perfect. Conversely, similar to the semi-supervised learning case [Dai et al., 2017], it can even benefit from an imperfect generator.

# 3  OUR APPROACH

MetaGAN is a conceptually simple and general framework for few-shot learning problems. Given a decent $K$-shot $N$-way classifier, similar to [Salimans et al., 2016] we introduce a conditional generative model with the objective to generate samples which are not distinguishable from true data sampled from a specific task. We increase the dimension of the classifier output from $N$ to $N + 1$, to model the probability that input data is fake. We train the discriminator (classifier) and generator in an adversarial setup.

The key idea behind MetaGAN is that imperfect generators in GAN models can provide fake data between the manifolds of different real data classes, thus providing additional training signals to the classifier as well as making the decision boundaries much sharper. We first describe our basic model formally in section 3.1, then introduce details of different instances of MetaGAN in following sections.

## 3.1  BASIC ALGORITHM

We first introduce the basic formulation of MetaGAN here. For a few-shot $N$-way classification problem $P(\mathcal{T})$ and dataset $\{\mathcal{T}_i\}_{i=1}^M$, assume we have one of the state-of-the-art few-shot classifiers $p_D(\mathbf{x}; \mathcal{T}) = (p_1(\mathbf{x}), p_2(\mathbf{x}), \cdots p_N(\mathbf{x}))$. Note that $D$ is conditioned on a specific task $\mathcal{T}$. In practice, this conditioning can be either via fast adaptation [Finn et al., 2017] or feeding the support set as input [Snell et al., 2017, Mishra et al., 2018, Sung et al., 2018]. We augment the classifier with an additional output, as done in semi-supervised learning with GANs [Salimans et al., 2016]: $p_D(\mathbf{x}; \mathcal{T}) = (p_1(\mathbf{x}), p_2(\mathbf{x}), \cdots p_N(\mathbf{x}), p_{N+1}(\mathbf{x}))$. We also train a task-conditioned generator $G(\mathbf{z}, \mathcal{T})$ with generating distribution $p_G^{\mathcal{T}}(\mathbf{x})$ that tries to generate data for the specific task $\mathcal{T}$. Then for the training episode of task $\mathcal{T}$ we maximize the following combination of the $N$-way classification objective and the real/fake classification objective for the discriminator:

$$\mathcal{L}_D^{\mathcal{T}} = \mathcal{L}_{\text{supervised}} + \mathcal{L}_{\text{unsupervised}}, \tag{1}$$

$$\mathcal{L}_{\text{supervised}} = \mathbb{E}_{\mathbf{x}, y \sim Q_{\mathcal{T}}^s} \log p_D(y | \mathbf{x}, y \leq N) \tag{2}$$

$$\mathcal{L}_{\text{unsupervised}} = \mathbb{E}_{\mathbf{x} \sim Q_{\mathcal{T}}^u} \log p_D(y \leq N | \mathbf{x}) + \mathbb{E}_{\mathbf{x} \sim p_G^{\mathcal{T}}} \log p_D(N + 1 | \mathbf{x}) \tag{3}$$

For the generator, we minimize the non-saturating generator loss

$$L_G^{\mathcal{T}}(D) = -\mathbb{E}_{\mathbf{x} \sim p_G^{\mathcal{T}}}[\log(p_D(y \leq N | \mathbf{x}))]. \tag{4}$$

Then the overall objective for training MetaGAN is

$$\mathcal{L}_D = \max_D \mathbb{E}_{\mathcal{T} \sim P(\mathcal{T})} \mathcal{L}_D^{\mathcal{T}} \tag{5}$$

$$\mathcal{L}_G = \min_G \mathbb{E}_{\mathcal{T} \sim P(\mathcal{T})} \mathcal{L}_G^{\mathcal{T}}. \tag{6}$$

## 3.2  DISCRIMINATOR

MetaGAN generally doesn't impose restrictions on the design of discriminator. It can be adapted from almost any state-of-the-art few-shot learners. We adopt two popular choices of few-shot classification models as our disciminator, MAML[Finn et al., 2017] and Relation Networks [Sung et al., 2018], representing learning to fast fine-tune based models and learning shared embedding and metric based models respectively.

### 3.2.1 METAGAN WITH MAML

MAML trains a transferable initialization that is able to quickly adapt to any specific task with one step gradient descent. Formally the discriminator $D(\theta_d)$ is parametrized by parameters $\theta_d$. For a specific task $\mathcal{T} \sim P(\mathcal{T})$, we update the parameters to $\theta_d' = \theta_d - \alpha \nabla_{\theta_d} \ell_D^{\mathcal{T}}$ according to the loss eq. 7

$$\ell_D^{\mathcal{T}} = -\mathbb{E}_{\mathbf{x},y \sim S_{\mathcal{T}}^s} \log p_D(y|\mathbf{x}, y \leq N) - \mathbb{E}_{\mathbf{x} \sim S_{\mathcal{T}}^u} \log p_D(y \leq N|\mathbf{x}) - \mathbb{E}_{\mathbf{x} \sim p_G^{\mathcal{T}}} \log p_D(N+1|\mathbf{x}). \quad (7)$$

Then we minimize the expected loss on query set with adapted discriminator $D(\theta_d')$ across tasks $\mathcal{T}$ to train the discriminator's initial parameters $\theta_d$, and we train the generator using adapted discriminator $D(\theta_d')$. Finally our whole model combining MetaGAN with MAML can be trained using the loss introduced in eq. 5 and eq. 6, as shown below:

$$\mathcal{L}_D = \max_D \mathbb{E}_{\mathcal{T} \sim P(\mathcal{T})} \mathcal{L}_{D(\theta_d')}^{\mathcal{T}} \quad (8)$$

$$\mathcal{L}_G = \min_G \mathbb{E}_{\mathcal{T} \sim P(\mathcal{T})} \mathcal{L}_G^{\mathcal{L}}(D(\theta_d')). \quad (9)$$

We put the detailed algorithms for training MetaGAN with MAML model in the supplemental material.

### 3.2.2 METAGAN WITH RELATION NETWORK

The Relation Network (RN) is a few-shot learning model aiming to do classification via learning a deep distance metric between images. MetaGAN can integrate with RN in a principled and straightforward way.

For a specific task $\mathcal{T} \sim P(\mathcal{T})$, following [Sung et al., 2018] let $r_{i,j} = g_\psi(\mathcal{C}(f_\phi(\mathbf{x}_i), f_\phi(\mathbf{x}_j))), \mathbf{x}_i \in S_{\mathcal{T}}^s, \mathbf{x}_j \in Q_{\mathcal{T}}^s$ be the relevance score between query set image $\mathbf{x}_j$ and support set image $\mathbf{x}_i$, where $g_\psi$ is the relation module, $f_\phi$ is the feature embedding network and $\mathcal{C}$ is the concatenation operator. Different from [Sung et al., 2018] we don't restrict $r_{i,j}$ to be in range of 0 to 1, we rather use $r_{i,j}$ as logits used in softmax classification

$$p_D(y = k|\mathbf{x}_j) = \frac{\exp(r_{k,j})}{1 + \sum_{i=1}^N \exp(r_{i,j})} \quad (10)$$

We adopt the simple trick proposed in [Salimans et al., 2016] by setting the logit of the fake class to 0, which is corresponding to the constant 1 appearing in denominator, to model $p_D(N+1|\mathbf{x})$ which is the probability that input data is fake. Thus we can train our model, MetaGAN with RN, directly using loss eq. 5 and eq. 6.

### 3.3 GENERATOR

We use a conditional generative model to generate fake data that is close to the real data manifold in one specific task $\mathcal{T}$. To do so, we first compress the information in the task's support dataset with a dataset encoder $E$ into vector $h_{\mathcal{T}}$, which contains sufficient statistics for the data distribution of task $\mathcal{T}$. Then $h_{\mathcal{T}}$ is concatenated with random noise input $z$ to be provided as input to the generator network. Inspired by the statistic network proposed in [Edwards and Storkey, 2017], our dataset encoder is composed of two modules:

**Instance-Encoder Module** The Instance-Encoder is a neural network that learns a feature representation for each individual data example in the dataset $S_{\mathcal{T}}^s$. It maps each data example $\mathbf{x}_i \in S_{\mathcal{T}}^s$ to feature space $e_i = \textit{Instance-Encoder}(\mathbf{x}_i)$.

**Feature-Aggregation Module** The Feature-Aggregation module takes each embedded feature vector $e_i$ as input and produce the representation vector $h_{\mathcal{T}}$ for the whole task training set. Feasible aggregation methods include average pooling, max pooling and other element-wise aggregation operators. We use average pooling following [Edwards and Storkey, 2017] in our MetaGAN model.

By integrating an Instance-Encoder module and a Feature-Aggregation Module, the instance-encoder is encouraged to learn a representation such that averaging different samples in the learned feature space makes sense. Also, feature-aggregation makes it harder for the generator to simply reconstruct its inputs, which can lead to mode dropping [Che et al., 2017].

### 3.4 LEARNING SETTINGS

In this section we show that both supervised few-shot learning and semi-supervised few-shot learning can be unified in the MetaGAN framework.

**Supervised Few-Shot Learning** Supervised learning is the most common learning setting of few-shot classification models. For a task $\mathcal{T} \sim P(\mathcal{T})$, since an unlabeled set $S_{\mathcal{T}}^u$ and $Q_{\mathcal{T}}^u$ is not available, we use the labeled set $S_{\mathcal{T}}^s$ and $Q_{\mathcal{T}}^s$ to replace them respectively in loss eq. 1 and eq. 7.

**Sample-Level Semi-Supervised Few-Shot Learning** Sample-level semi-supervised learning follows the same setup as [Ren et al., 2018], where unlabeled data examples are available in each task. While our model is flexible enough to deal with different sets of unlabeled examples in the support set and the query set, for a task $\mathcal{T} \sim P(\mathcal{T})$ we only use a single unlabeled set of examples $U_{\mathcal{T}}$ to follow the same training scheme in [Ren et al., 2018], for a better comparison with our baseline models.

Specifically, for MetaGAN with MAML, we set $S_{\mathcal{T}}^u = S_{\mathcal{T}}^s$ and $Q_{\mathcal{T}}^u = U_{\mathcal{T}}$. For MetaGAN with RN, we set $S_{\mathcal{T}}^u = \emptyset$ and $Q_{\mathcal{T}}^u = U_{\mathcal{T}}$ in loss eq. 1 and eq. 7.

**Task-Level Semi-Supervised Few-Shot Learning** For Task-level semi-supervised learning, the training dataset $\{\mathcal{T}_i\}_{i=1}^M$ consisting of labeled tasks and unlabeled tasks. For labeled tasks we simply follow the supervised learning setting described above. For unlabeled tasks, we omit the supervised loss term by setting $Q_{\mathcal{T}}^s = \emptyset$ and $S_{\mathcal{T}}^s = \emptyset$ in loss eq. 1 and eq. 7.

As proposed in [Salimans et al., 2016] we adopt the "feature matching loss" as the generator loss $\mathcal{L}_G$ in both sample-level and task-level semi-supervised few-shot learning.

## 4 WHY DOES METAGAN WORK?

In this section, we introduce intuition as well as theoretical justifications of MetaGAN, which motivate various improvements we made on the model.

In a few-shot classification problem, the model tries to optimize a decision boundary for each task with just a few samples in each class. Obviously this problem is impossible if no information can be learned from other tasks, as there are so many possible decision boundaries to separate the few samples apart and most of them will not generalize. Meta-learning tries to learn a shared strategy across different tasks to form decision boundaries from few samples, in the hope that this strategy is able to generalize to new tasks.

Although this is reasonable, there can be some problems. For example, some objects look more similar than others. It may be easier to form a decision boundary between a cat and a car than between a cat and a dog. If the training data does not contain tasks that try to separate a cat and a dog, it may feels difficult to extract the correct features to separate these two classes of objects. However, on the other hand, the expectation to have all kinds of class combinations during training leads to the combinatorial explosion problem.

This is where our proposed MetaGAN formulation helps. Just as for the case of doing semi-supervised learning with GANs, we don't expect our generator to generate data that is exactly on the true data manifold. Instead, it is better that the generator is able to generate data a bit off the data manifold of each class, cf. fig. 1. This forces our discriminator to learn a much sharper decision boundary. Instead of only learning to separate cats and dogs, the discriminator of MetaGAN is forced to learn not only what are real cats or dogs, but also what are fake data generated from where is a bit off the cat and dog manifold. The discriminator thus has to extract features strong enough to decide the boundary of the real data manifold, which helps to separate different classes apart. Moreover, the separation between real/fake classes is independent of the class combinations selected during the few-shot learning process.

Following the ideas behind the theoretical justifications studied in the semi-supervised learning setting, we provide similar justifications in the few-shot learning problem. We include the formal statement of the assumptions in the supplemental material.

First, as in [Dai et al., 2017], for a specific task $\mathcal{T}$, we assume that the classifier relies on a feature extractor $f_{\mathcal{T}}$ to perform classification. We also make the assumption that $G(\cdot; \mathcal{T})$ is a "separating complement generator" (which we define in the supplemental material) for each task $\mathcal{T}$. Intuitively this means that the generator $G(z; \mathcal{T})$ satisfies two conditions: 1) the generator distribution $p_G^{\mathcal{T}}$ has a

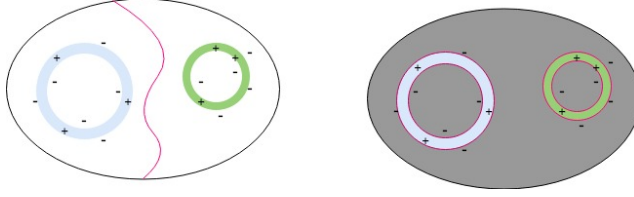

Figure 1: Left: decision boundary without metaGAN. Right: decision boundary with metaGAN. We use red curves to denote the decision boundary. Blue area in figure represents class A, green area represents class B, and gray area represents fake class. We use $+$ to denote real samples and $-$ to denote fake samples generated.

high density region that is disjoint with the data manifold of all classes; 2) This high density region of $p_G^{\mathcal{T}}$ can separate manifolds of different classes.

Then by following arguments similar to those in [Dai et al., 2017], we can prove the following:

**Theorem 1** *Let $G_{\mathcal{T}}$ be a separating complement generator in each task $\mathcal{T}$ sampled from $P(\mathcal{T})$. Denote $S_{\mathcal{T}}$ the support set and $F_{\mathcal{T}}$ the generated fake dataset. We assume our learned meta-learner is able to learn a classifier $D_{\mathcal{T}}$ which obtains a strong correct decision boundary on the augmented support set($S_{\mathcal{T}}, F_{\mathcal{T}}$). Then if $|F_{\mathcal{T}}| \to +\infty$, then $D_{\mathcal{T}}$ can almost surely correctly classify all real samples from the data distribution $p_{\mathcal{T}}(x)$ of the task.*

The theorem is saying that if we have a generator that is neither too good nor too bad, but can generate data around the the real class manifold and have a high density region that can help separating different classes apart, then the generated data together with a few real data can help us determine the correct decision boundary.

## 5 EXPERIMENTS

### 5.1 DATASETS

**Omniglot** is a dataset consisting of handwritten character images from 50 languages. There are 1623 classes of characters with 20 examples within each class. Following prior training and the evaluation protocol used in [Vinyals et al., 2016], we downsampled all images to $28 \times 28$ and randomly split the dataset into 1200 classes for traininig and 432 classes for testing. The same data augmentation techniques proposed by [Santoro et al., 2016] are utilized, randomly rotating each image by a multiple of 90 degrees to form new classes.

**Mini-Imagenet** is a modified subset of the well-known ILSVRC-12 dataset, consisting of $84 \times 84$ colored images from 100 classes with 600 random samples in each class. We follow the same class split as in [Ravi and Larochelle, 2017], that takes 64 classes for training, 16 classes for validation and 20 classes for testing.

### 5.2 SUPERVISED FEW-SHOT LEARNING

On the Omniglot dataset, MetaGAN with MAML shares the same discriminator network architecture and most model hyper-parameters setup with vanilla convolutional MAML[Finn et al., 2017]. We set the meta batch-size to 16 for 5-way classification and 8 for 20-way classification to fit the memory limit of the GPU. For MetaGAN with RN, we batch 15 query images for each class for both 1-shot 5-way and 5-shot 5-way classification, and we batch 5 query images for each class for 1-shot 20-way and 5-shot 20-way task. We set the meta batch-size of MetaGAN with RN model to 1 in our all experiments.

On Mini-Imagenet dataset, we train our MetaGAN with the MAML model using the first-order approximation method with 1 gradient step as proposed in [Finn et al., 2017], due to the consideration of computational cost.

For the conditional generator we adopt a ResNet-like architecture inspired by [Gulrajani et al., 2017] in both models; see more details of the architecture of the generator in supplemental material.

|  | 5-way Acc. | | 20-way Acc. | |
| Model | 1-shot | 5-shot | 1-shot | 5-shot |
| --- | --- | --- | --- | --- |
| Neural Statistician | 98.1 | 99.5 | 93.2 | 98.1 |
| Prototypical Nets | 98.8 | 99.7 | 96.0 | 98.9 |
| MAML | $98.7 \pm 0.4$ | $\mathbf{99.9 \pm 0.1}$ | $95.8 \pm 0.3$ | $98.9 \pm 0.2$ |
| Ours: MetaGAN + MAML | $99.1 \pm 0.3$ | $99.7 \pm 0.21$ | $96.4 \pm 0.27$ | $98.9 \pm 0.18$ |
| Relation Net | $99.6 \pm 0.2$ | $99.8 \pm 0.1$ | $97.6 \pm 0.2$ | $99.1 \pm 0.1$ |
| Ours: MetaGAN + RN | $\mathbf{99.67 \pm 0.18}$ | $99.86 \pm 0.11$ | $\mathbf{97.64 \pm 0.17}$ | $\mathbf{99.21 \pm 0.1}$ |

Table 1: Few-shot classification results on Omniglot.

|  | 5-way Acc. | |
| Model | 1-shot | 5-shot |
| --- | --- | --- |
| Prototypical Nets | $49.42 \pm 0.78$ | $68.20 \pm 0.66$ |
| MAML(5 gradient steps) | $48.70 \pm 1.84$ | $63.11 \pm 0.92$ |
| MAML(5 gradient steps, first order) | $48.07 \pm 1.75$ | $63.15 \pm 0.91$ |
| MAML(1 gradient step, first order) | $43.64 \pm 1.91$ | $58.72 \pm 1.20$ |
| Ours: MetaGAN + MAML(1 step, first order) | $46.13 \pm 1.78$ | $60.71 \pm 0.89$ |
| Relation Net | $50.44 \pm 0.82$ | $65.32 \pm 0.7$ |
| Ours: MetaGAN + RN | $\mathbf{52.71 \pm 0.64}$ | $\mathbf{68.63 \pm 0.67}$ |

Table 2: Few-shot classification results on Mini-Imagenet.

We use the Adam [Kingma and Ba, 2014] optimizer with initial learning rate as 0.001, $\beta_1 = 0.5$ and $\beta_2 = 0.9$ to train both generator and discriminator networks. For Omniglot we decay the learning rate starting from 10K batch updates, and cut it in half for every 10K following updates. For Mini-Imagenet we decay the learning rate starting from 30K batch updates, and cut it in half for every 10K updates.

We present our results of 5-way and 20-way few-shot classification for Omniglot dataset in table 1, and show results of Mini-Imagenet dataset in table 2. We see that our proposed MetaGAN consistently improves over baseline classifiers, and achieves comparable or outperforms state-of-the-art performance on the challenging Mini-Imagenet benchmark.

## 5.3 SAMPLE-LEVEL SEMI-SUPERVISED FEW-SHOT LEARNING

As introduced in section 3.4, we evaluate the effectiveness of our proposed MetaGAN in the sample-level semi-supervised few-shot learning setting, following a similar training and evaluation scheme without "distractors" to that proposed in [Ren et al., 2018] (We will point out the differences in the scheme later on). For the Omniglot dataset we sample 10% of the images of each class to form the labeled set, and take all remaining data as the unlabeled set. For Mini-Imagenet we sample 40% images of each class as the labeled set, and sample 5 images of each class for each training episode.

Note that our model only leverages unlabeled samples during the training phase, while the refining model proposed in [Ren et al., 2018] uses unlabeled samples in both training (5 samples for each class) and evaluation phases (20 samples for each class). This makes our model acquire strictly less information during evaluation, compared to [Ren et al., 2018]. The classifier trained with our proposed MetaGAN formulation is encouraged to form better decision boundaries by utilizing unlabeled and fake data, and is free from the demands of unlabeled samples during testing, different from the kmeans-based refining model [Ren et al., 2018] which strongly relies on the unlabeled data for testing.

| | Omniglot | Mini-Imagenet | |
| Model | 1-shot 5-way | 1-shot 5-way | 5-shot 5-way |
|---|---|---|---|
| Prototypical Nets(Supervised) | $94.62 \pm 0.09$ | $43.61 \pm 0.27$ | $59.08 \pm 0.22$ |
| Semi-Supervised Inference(PN) | $97.45 \pm 0.05$ | $48.98 \pm 0.34$ | $63.77 \pm 0.20$ |
| Soft k-Means | $97.25 \pm 0.10$ | $50.09 \pm 0.45$ | $\mathbf{64.59 \pm 0.28}$ |
| Soft k-Means+Cluster | $\mathbf{97.68 \pm 0.07}$ | $49.03 \pm 0.24$ | $63.08 \pm 0.18$ |
| Masked Soft k-Means | $97.52 \pm 0.07$ | $\mathbf{50.41 \pm 0.31}$ | $64.39 \pm 0.24$ |
| Ours: Relation Nets(Supervised) | $94.81 \pm 0.08$ | $44.24 \pm 0.24$ | $58.72 \pm 0.31$ |
| Ours: MetaGAN + RN | $97.58 \pm 0.07$ | $50.35 \pm 0.23$ | $64.43 \pm 0.27$ |

Table 3: Sample-level Semi-Supervised Few-shot classification results on Omniglot and Mini-Imagenet.

| | Omniglot | Mini-Imagenet |
| Model | 1-shot 5-way | 1-shot 5-way |
|---|---|---|
| Prototypical Net(Supervised) | $93.66 \pm 0.09$ | $42.28 \pm 0.32$ |
| Relation Net(Supervised) | $93.82 \pm 0.07$ | $43.87 \pm 0.20$ |
| Ours: MetaGAN + RN | $\mathbf{97.12 \pm 0.08}$ | $\mathbf{47.43 \pm 0.27}$ |

Table 4: Task-level Semi-Supervised 1-shot classification results on Omniglot and Mini-Imagenet.

We display the results of sample-level semi-supervised few-shot classification results on Omniglot and Mini-Imagenet in table 3. Though our model cannot be compared with the kmeans refining model directly as discussed above, we obtain comparable state-of-the-art results on both 1-shot and 5-shot tasks, while significantly improving the purely supervised baseline models.

## 5.4 TASK-LEVEL SEMI-SUPERVISED FEW-SHOT LEARNING

We proposed a new learning setting for the few-shot learning problem in section 3.4: task-level semi-supervised few-shot learning. In this learning setting, existing few-shot learning models[Ravi and Larochelle, 2017, Sung et al., 2018, Ren et al., 2018] are unable to effectively leverage purely unsupervised tasks, which consist of only unlabeled samples in both support set and query set.

To demonstrate that our proposed MetaGAN model can successfully learn from unsupervised tasks, we create new splits of Omniglot and Mini-Imagenet datasets. For the Omniglot dataset we randomly sample 10% of classes from the training set as a labeled set of classes, and the remaining 90% classes as an unlabeled set of classes. For Mini-Imagenet dataset we randomly sample 40% as labeled classes and the remaining 60% are unlabeled. The validation set and test set of each dataset remains unchanged, using all classes to evaluate the performance of models. During training time, we sample supervised tasks only from the labeled set of classes, and sample unsupervised tasks from the unlabeled set of classes. We alternate between sampled supervised tasks and sampled unsupervised tasks for training the MetaGAN model, while we only use sampled supervised tasks to train the baseline model.

We show the results of task-level semi-supervised few-shot classification results on Omniglot and Mini-Imagenet in table 4. By integrating the baseline model into the MetaGAN framework, the model effectively learned to utilize the unsupervised tasks for helping the classification task, showing that MetaGAN can learn transferable knowledge from totally unsupervised tasks.

## 6 CONCLUSION

We propose MetaGAN, a simple and generic framework to boost the performance of few-shot learning models. Our approach is based on the idea that fake samples produced by the generator can help classifiers learn a sharper decision boundary between different classes from a few samples.

We make an analogy between few-shot learning and semi-supervised learning- both of them have only a few labeled data and both can benefit from an imperfect generator. Then we modified the techniques used for semi-supervised learning with GANs to work in the few-shot learning scenario. We give intuitive as well as theoretical justifications of the proposed approach.

We demonstrated the strength of our algorithm on a series of few-shot learning and semi-supervised few-shot learning tasks. For future work, we plan to extend MetaGAN to the few-shot imitation learning setting.

## ACKNOWLEDGEMENT

We thank Intel Corporation for supporting our deep learning related research.

## Footnotes

[3]However, it is worth noticing that MetaGAN can also be easily combined with other models, such as prototypical networks or SNAIL.

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
