[Supplementary Material · supplementary.pdf]

# Supplemental Material

## 1  THEORETICAL JUSTIFICATION

In this section we define the conditions and prove the theorem stated in the main text. Techniques are similar to [1].

Recall that we say a measurable set $U$ is a support set of probability measure $p$ iff $p(U) = 1$. A measurable set $U$ is called $p$-dense iff for every point $z \in U$, for any neighborhood $U(z, \epsilon)$ of $z$, we have $p(U(z, \epsilon)) > 0$.

Fixing a task $\mathcal{T}$, our discriminator first compresses each data point $\mathbf{x}$ to a latent vector representation $\mathbf{z} = f(\mathbf{x})$, and then pass to a linear classifier, with weights $\mathbf{w}_i, i = 1, 2 \cdots K$. We further assume $\mathbf{w}_i, i = 1, 2 \cdots K$ are bounded by a uniform constant $C$.

we denote $p_{\mathcal{T}}(\mathbf{z})$ the distribution of latent representations of task $\mathcal{T}$. We assume that $p_{\mathcal{T}}$ has compact support $B$. Without loss of generality, we can also assume $B$ is convex, otherwise we can take its convex closure. We also denote the probability distribution of latent distributions of class $i$ as $p_{\mathcal{T}}^i(\mathbf{z}), i = 1 \ldots K$. We define an open domain $U \subset \mathbb{R}^n$ is an $\epsilon$-support of a probability measure $p$, if $p(U) > 1 - \epsilon$. We assume that there exists some very small $\epsilon > 0$, we have a set of $U_i, i = 1, 2 \cdots, K$, such that $U_i$ is an $\epsilon$-support of $p_{\mathcal{T}}^i$ for all $i = 1, 2, \cdots K$. We also assume all $U_i$ is disjoint from each other. For the adapted generator $G_{\mathcal{T}}(\mathbf{z})$, we denote its corresponding distribution in latent space as $p_{\mathcal{T}}^G(\mathbf{z})$. Assume $p_{\mathcal{T}}^G$ has $p_{\mathcal{T}}^G$-dense set $S_G \subset B$.

Now we can define what is a "complement separating generator".

**Definition 1.** *With the above assumptions and notations, we call a generator $G(z; \cdot)$ a complement separating generator if, for any task $\mathcal{T} \sim p_{\mathcal{T}}$, $G(z; \mathcal{T})$ satisfies the following two conditions:*

- *for all $i = 1, 2, \cdots K$, $U_i \cap S_G = \emptyset$.*

- *for all $i, j = 1, 2, \cdots K$, $U_i$ and $U_j$ are pathwise disconnected from each other in $B \setminus S_G$.*

Then we can formally state the main theorem as:

**Theorem 1.** *Let $G_{\mathcal{T}}$ be a separating complement generator. Denote $S_{\mathcal{T}}$ the support(training) set and $F_{\mathcal{T}}$ the generated fake dataset. We assume our learned meta-learner is able to learn a classifier $D_{\mathcal{T}}$ which obtains strong correct decision boundary on the augmented support set $(S_{\mathcal{T}}, F_{\mathcal{T}})$. More precisely, (1) for $\mathbf{x}, y \in S_{\mathcal{T}}$, $\mathbf{x} \cdot \mathbf{w}_y > \max\{0, \mathbf{x} \cdot \mathbf{w}_i\}$ for all $i \neq y$. (2) for $f(\mathbf{x}) \in F_{\mathcal{T}}$, $f(\mathbf{x}) \cdot \mathbf{w}_i < 0$ for all $i \leq K$.*

*Then if $|F_{\mathcal{T}}| \to +\infty$, then $D_{\mathcal{T}}$ can almost surely correctly classify all real samples from the data distribution $p_{\mathcal{T}}(x)$ of the task.*

*Proof.* We first need to prove when $|F_{\mathcal{T}}| \to +\infty$, for all $\mathbf{z} \in S_G$, we have almost surely $\max_{i \leq K} \mathbf{w}_i \cdot \mathbf{z} \leq 0$. The detailed proof is subtle. Here we only give a sketch. From the assumption that $S_G$ is $p_{\mathcal{T}}^G$-dense, one can easily deduce that when $|F_{\mathcal{T}}| \to +\infty$, the points $F_{\mathcal{T}}$ become dense in $S_G$. More precisely, for any $\epsilon > 0$, any $\mathbf{z} \in S_G$, when $|F_{\mathcal{T}}| \to +\infty$, then almost surely there exits $\mathbf{z}' \in F_{\mathcal{T}}$, such that $|\mathbf{z} - \mathbf{z}'| < \epsilon$. From the assumption $\mathbf{w}_i, i = 1, 2 \cdots K$ are bounded by a uniform constant $C$, we can get almost surely $\max_{i \leq K} \mathbf{w}_i \cdot \mathbf{z} \leq 0$.

Then we prove by contradiction. If for any task $\mathcal{T}$, $D$ successfully adapted to a support set $(S_{\mathcal{T}}, F_{\mathcal{T}})$, without loss of generality, we can assume $S_{\mathcal{T}} = \{(\mathbf{x}_i, y_i)\}_{i=1}^{K}$ is one-shot. If there is a data

point $(\mathbf{x}, y)$ which is classifier incorrectly, namely there exists some $j \neq y$, such that $\mathbf{w}_j \cdot f(\mathbf{x}) > \mathbf{w}_y \cdot f(\mathbf{x}) > 0$. In the mean time $\mathbf{w}_j \cdot f(\mathbf{x}_j) > 0$. So for all $\alpha \in [0,1]$, $\mathbf{w}_j \cdot [\alpha f(\mathbf{x}_j) + (1-\alpha)f(\mathbf{x})] > 0$. This contradicts with two facts: 1) $U_y$ and $U_j$ are pathwise disconnected from each other in $B \setminus S_G$; 2) almost surely $\max_{i \leq K} \mathbf{w}_i \cdot \mathbf{z} \leq 0$, for all $\mathbf{z} \in S_G$.

So the theorem is proved. $\qquad\qquad\qquad\qquad\qquad\qquad\qquad\qquad\qquad\qquad\qquad\square$

## 2  ALGORITHMS FOR TRAINING METAGAN WITH MAML

We describe the detailed algorithm for training MetaGAN with MAML model as following:

---
**Algorithm 1** MetaGAN with MAML
---
$G(\mathbf{z}, \mathcal{T})$**: Generator network parameterized by** $\theta_g$.
$D(x)$**: Discriminator network. parameterized by** $\theta_d$.

   Initialize $\theta_g, \theta_d$ randomly.
   **while** not done **do**
      Sample a batch of tasks $\mathcal{T}_i \sim p(\mathcal{T})$.                    $\triangleright$ Discriminator Update
      **for** all $\mathcal{T}_i$ **do**
         Get $K$ real samples $\mathcal{D}_r = \{\mathbf{x}^{(i)}, y^{(i)}\}$ from $\mathcal{T}_i$.
         Sample $K$ generated samples $\mathcal{D}_f = \{\mathbf{x}^{(j)}\} = G(\mathbf{z}^{(j)}, \mathcal{T}_i)$ from $G(\mathbf{z}, \mathcal{T}_i)$.
         Evaluate discriminator loss $\ell_D^{\mathcal{T}_i}$ with $D_r$ and $D_f$.
         Compute adapted discriminator parameters $\theta'_{d_i} = \theta_d - \alpha \nabla_{\theta_d} \ell_D^{\mathcal{T}_i}$.
      **end for**
      Update $\theta_d$ using loss $\mathcal{L}_D$
      Sample a batch of tasks $\mathcal{T}_i \sim p(\mathcal{T})$.                    $\triangleright$ Generator Update
      **for** all $\mathcal{T}_i$ **do**
         Sample $K$ generated samples $\mathcal{D}_f = \{\mathbf{x}^{(j)} = G(\mathbf{z}^{(j)}, \mathcal{T}_i)\}$ from $G(\mathbf{z}, \mathcal{T}_i)$.
         Compute adapted discriminator parameters $\theta'_{d_i} = \theta_d - \alpha \nabla_{\theta_d} L_D$.
         Compute generator loss gradient $\nabla_{\theta_g} L_G^{\mathcal{T}_i}$ with the adapted discriminator.
      **end for**
      Update generator parameters $\theta_g$ with accumulated generator loss gradients.
   **end while**

---

## 3  GENERATOR AND DISCRIMINATOR ARCHITECTURE

### 3.1  GENERATOR

We describe the generator architecture used in Omniglot models in table 3.2. The generator used in Mini-Imagenet models are similar. Please refer to provided code [1] for more details on the network architecture and training hyperparameters.

### 3.2  DISCRIMINATOR

For both model MetaGAN with MAML and MetaGAN with RN, we adopt the same neural network architecture as MAML and RN respectively.

| |
|---|
| $2\times$ { *conv2d* 64 feature maps with $3 \times 3$ kernels and Leaky-Relu activations } |
| *conv2d* 64 feature maps with $3 \times 3$ kernels, stride 2 and Leaky-Relu activations |
| $2\times$ {*conv2d* 128 feature maps with $3 \times 3$ kernels and Leaky-Relu activations } |
| *conv2d* 128 feature maps with $3 \times 3$ kernels, stride 2 and Leaky-Relu activations |
| $2\times$ { *conv2d* 256 feature maps with $3 \times 3$ kernels and Leaky-Relu activations } |
| *conv2d* 256 feature maps with $3 \times 3$ kernels, stride 2 and Leaky-Relu activations |
| *fully-connected* layer with 256 units and Leaky-Relu activations |
| *sample-dropout* and *concatenation* with number of samples |
| *average pooling* within each dataset |
| *concatenation* embeded features with noise input $z$ |
| *upsample conv2d* 512 feature maps with $3 \times 3$ kernels and Leaky-Relu activations with residual connection |
| *upsample conv2d* 256 feature maps with $3 \times 3$ kernels and Leaky-Relu activations with residual connection |
| *upsample conv2d* 128 feature maps with $3 \times 3$ kernels and Leaky-Relu activations with residual connection |
| *upsample conv2d* 1 feature maps with $3 \times 3$ kernels and Leaky-Relu activations with residual connection |

Table 1: Omniglot Conditional Generator

## Footnotes

[1]https://github.com/sodabeta7/MetaGAN

## References

[1] Zihang Dai, Zhilin Yang, Fan Yang, William W Cohen, and Ruslan R Salakhutdinov. Good semi-supervised learning that requires a bad gan. In I. Guyon, U. V. Luxburg, S. Bengio, H. Wallach, R. Fergus, S. Vishwanathan, and R. Garnett, editors, *Advances in Neural Information Processing Systems 30*, pages 6510–6520. Curran Associates, Inc., 2017.