[Reviews · NeurIPS 2018]

Reviewer 1



This paper proposes a method of improving upon existing meta-learning approaches by augmenting the training with a GAN setup. The basic idea has been explored in the context of semi-supervised learning: add an additional class to the classifier's outputs and train the classifier/discriminator to classify generated data as this additional fake class. This paper extends the reasoning for why it might work for semi supervised learning to why is might work for few-shot meta learning. The clarity of this paper could be greatly improved. They are presenting many different variants of few-shot learning in supervised and semi-supervised setting, and the notation is a bit tricky to follow initially. The idea proposed in this paper is a very simple one, yet the paper was a bit laborious to read. Overall, this paper on the border of acceptance. The idea is simple, and although it has been used in other contexts, its application in this area is novel. However, since the basic ideas in this paper a not new, and rather just a combination of existing work, I would hope for a bit more empirical evaluations (for example, in the RL setting as MAML is applied). That said, the method is simple and improves upon existing approaches and so I am (very marginally) on the positive side of acceptance boundary, provided the paper could edited for clarity. Detailed comments: - The notation in "few shot learning section" is a bit hard to read. For example, the support set is given as S_T = S^s_T U S^u_T and then S^s_T is described as the supervised support set, so I would assume S^u_T is the unsupervised support set. But then U_T is defined as the unlabelled support set , so what is S^u_T? Also, what do the *'s on line 71 mean? Why are some of the Y's *-ed and others not? In general, this paragraph could use some editing. - In table 2, why do you only apply the metaGAN framework to MAML with 1 step, first order? Why not the other versions, which are better than that one? - how significant are the omniglot improvements. Most gains look to be within error bars. - could the metaGAN approach be combed with the k-means method in table 3 in order to improve upon it?

Reviewer 2



POST-REBUTTAL UPDATE: The rebuttal addresses most of the issues addressed in the reviews, assuming that all promised changes will actually be incorporated in the camera-ready version. ----------------------------------------- The submission develops a meta-framework for few-shot learning that, building upon an approach/classifier of choice, uses a GAN approach to perform a kind of data augmentation. The classifier (discriminator) in this framework now has to not only distinguish the task classes, but also whether a sample was generated or not (similar to Dai et al.). Optimizing the overall objective yields improved results on both sample and task-based few shot learning problems. The overall idea and approach is quite sound, and the evaluation, demonstrating that the proposed meta-algorithm is applicable to various discriminator designs, is fairly convincing. The addition of the task-level semi-supervised setting is appreciated. However, it would be great to see slightly more extensive experiments on the other settings: compared to Ren et al., there is no evaluation on the distractor classes setting (l.39, cf. Ren et al.), nor on the tieredImageNet data set. Other remarks: - The submission is generally reasonably clear. - Section 2.1 is a bit to terse to read well. Several parts of the notation remain undefined. (X x Y, S_T^S, S_T^U, Q_T^S, Q_T_U, ...). - l.59-60: A word seems to be missing here. - l.123, l.127, l.141, and possible other places: Please refer to equations by "Eq. 5 and Eq. 6", etc., and not just writing the number. - Supplemental material, Sec. 3.2: I assume the table on the following page should either be embedded or referenced here.

Reviewer 3



This paper explores the idea of solving classification problems by first modeling the data generating distribution. While typical discriminative classifiers model p(class|data), generative models attempt to model the joint distribution p(class,data). The proposed MetaGAN tries to learn p(data), as part of a larger learning task trying to learn p(c|data). The key novelty in the paper is the insight that if you: * use GANs to capture the data distribution; * augment your classification problem with a "fake" class; * train a classifier simultaneously with a GAN; then the GAN can help the classifier learn better decision boundaries by helping to refine the boundary between true data distributions and the artificial fake class. The primary technical contribution of the paper is the combined loss function that blends all of these elements together. This seems like an entirely reasonable idea. However, I have several reservations about the paper: 1) I'm concerned about the loss function. The loss function described in Eqs (1)-(6) is based on simple maximum likelihood learning. However, there is a lot of research into the problems with these sorts of loss functions; significant effort has gone into alternative loss functions that typically perform much better in practice (see, eg, Wasserstein GANs, etc.). Why should we have confidence in this particular loss function? 2) The experiments aren't compelling. There are so many DNN papers out there, and so many different ideas, that the bar for publication (especially at NIPS) is very high. While your method demonstrates slight improvements on classification tasks, the tasks are not particularly state-of-the-art, and the algorithms compared against also don't feel like the state-of-the-art. This leads me to wonder: why should we publish this paper, as opposed to any of another million ideas? I think that ultimately this paper is solidly borderline - some good ideas, and a reasonable execution of those ideas, but with nothing particularly compelling.